A remaining useful life estimation method based on long short-term memory and federated learning for electric vehicles in smart cities

Chen Xuejiao 1
Chen Zhaonan 2
Zhang Mu 2 45380597@qq.com
Wang Zixuan 3
Liu Minyao 3
Fu Mengyi 3
Wang Pan 3
1 School of Communications, Nanjing Vocational College of Information Technology , Nanjing, Jiangsu , China
2 School of Mathematics and Information Science, Guiyang University , Guiyang, Guizhou , China
3 School of Modern Posts, Nanjing University of Posts and Telecommunications , Nanjing, Jiangsu , China
Aurangzeb Khursheed
Electronic publication date: 2023 Oct 25
Publication date: 2023
Volume: 9
Electronic Location ID: e1652
Received 2023 Jan 19; Accepted 2023 Sep 23
Copyright: © 2023 Chen et al.
Copyright year: 2023
Copyright holder: Chen et al.
License: This is an open access article distributed under the terms of the Creative Commons Attribution License, which permits unrestricted use, distribution, reproduction and adaptation in any medium and for any purpose provided that it is properly attributed. For attribution, the original author(s), title, publication source (PeerJ Computer Science) and either DOI or URL of the article must be cited.
License URL: https://creativecommons.org/licenses/by/4.0/

Keywords: Remaining useful life, Federated learning, Deep learning, Electric vehicles, Smart city, LSTM, State of health, New energy, Data privacy, Energy forecast

Funding: National Natural Science Foundation (General Program) China 61972211 Future Network Innovation Research and Application Projects 2021FNA02006 2021 Jiangsu Postgraduate Research Innovation Plan KYCX210794 The article is sponsored by the National Natural Science Foundation (General Program) Grant 61972211, China, the Future Network Innovation Research and Application Projects No. 2021FNA02006 and the 2021 Jiangsu Postgraduate Research Innovation Plan under Grant No. KYCX210794. The funders had no role in study design, data collection and analysis, decision to publish, or preparation of the manuscript.

==============================
In modern society, environmental sustainability is a top priority as one of the most promising entities in the new energy sector. Electric vehicles (EVs) are rapidly gaining popularity due to their promise of better performance and comfort. Above all, they can help address the problem of urban air pollution. Nonetheless, lithium batteries, one of the most essential and expensive components of EVs, have posed challenges, such as battery aging, personal safety, and recycling. Precisely estimating the remaining useful life (RUL) of lithium battery packs can effectively assist in enhancing the personal safety of EVs and facilitating secondary trading and recycling in other industries without compromising safety and reliability. However, the RUL estimation of batteries involves many variables, and the operating conditions of EV batteries are highly dynamic as they change with the environment and the driving style of the users. Many existing methods exist to estimate the RUL based on batteries’ state of health (SOH), but only some are suitable for real-world data. There are several difficulties as follows. Firstly, obtaining data about battery usage in the real world takes work. Secondly, most of these estimation models must be more representative and generalized because they are trained on separate data for each battery. Lastly, collecting data for centralized training may lead to a breach of user privacy. In this article, we propose an RUL estimation method utilizing a deep learning (DL) approach based on long short-term memory (LSTM) and federated learning (FL) to predict the RUL of lithium batteries. We refrain from incorporating unmeasurable variables as inputs and instead develop an estimation model leveraging LSTM, capitalizing on its ability to predict time series data. In addition, we apply the FL framework to train the model to protect users’ battery data privacy. We verified the results of the model on experimental data. Meanwhile, we analyzed the model on actual data by comparing its mean absolute and relative errors. The comparison of the training and prediction results of the three sets of experiments shows that the federated training method achieves higher accuracy in predicting battery RUL compared to the centralized training method and another DL method, with solid training stability.

Introduction

In modern society, environmental sustainability is always a top priority. In achieving sustainable development goals, the role of new energy sources has progressively grown in importance, contributing significantly to reducing carbon emissions (Gu & Liu, 2021). As one of the most promising entities in the new energy sector, electric vehicles (EVs) are rapidly gaining popularity due to their promise of better performance and comfort. Above all, they can help address the problem of urban air pollution.

The pivotal element of an EV resides in its lithium battery, which boasts ecological friendliness, extended longevity, and remarkable reliability in contrast to conventional battery types like lead-acid or NiMH batteries (Hannan et al., 2017; Zhang et al., 2019). Because of these advantages, lithium batteries are widely used in EVs and other critical applications, such as space applications, aircraft (Liu, Zhou & Peng, 2014), and backup energy systems.

Although lithium batteries are widely used, their failure can also be fatal. For example, in 2013, several Boeing 787s suffered fires due to lithium battery failures (Williard et al., 2013), while many car manufacturers recalled EVs due to fire safety concerns (Hawkins, 2020). Another issue with lithium batteries is cost. EVs are promising in many ways, but their high sale price remains a significant drawback (Bilgin et al., 2015), and lithium batteries, one of the most expensive components of EVs, are another major drawback (Andwari et al., 2017).

An accurate estimate of the remaining useful life (RUL) of a lithium battery pack can improve the personal safety of electric vehicles and allow owners to trade up and reduce costs without sacrificing safety or reliability. Currently, existing approaches to battery life prediction typically fall into two categories: model-based and data-driven methods. However, these approaches have limitations in predicting battery life in electric vehicles, as they require either extensive physical knowledge or large amounts of experimental data for model-based approaches, or complex and uncertain condition monitoring data for data-driven strategies. Moreover, the relevant data used in these approaches is not readily available on electric vehicles, and the data used for model training is obtained in the laboratory, which cannot be generalized to realistic situations.

Therefore, researchers have introduced data-driven deep learning (DL) based methods to study battery RUL as an alternative to model-based approaches. DL does not use human-designed features. Instead, the model automatically extracts complex structural features by training multiple non-linear networks with strong generalization capability.

A traditional deep learning-based model for RUL prediction, which transfers battery data from electric vehicles to a cloud server for centralized training, has the following drawbacks: 1) Data privacy issues. As big data develops and users become more aware of security and privacy, there is an increased risk of privacy leakage (Lohiya & Thakkar, 2020).

2) Incomplete data distribution. Individual EV battery data rarely reflects how the battery is consumed and how long it will last under different scenarios. These problems result in insufficient training data for the model, negatively affecting its accuracy and reliability.

Google initially introduced Federated Learning (FL) in 2017 (McMahan et al., 2017). Its distinctive training methods make it a significant form of distributed learning, including consolidating model parameters and implementing data constraints on the device. FL serves as an effective solution to the challenge of data protection. By leveraging shared information and global prediction models, it holds the potential to enhance the precision of forecasting remaining battery energy in EVs.

This article proposes an RUL prediction model grounded in FL and powered by recurrent neural networks. Through the utilization of local models on EVs, the uploading of model updates helps circumvent privacy breaches that may occur during transmission. In addition, by incorporating EVs as sub-nodes, each node in the FL network will possess distinct driving data from EVs. Consequently, this augmented data distribution contributes to a more comprehensive training of the global model. The global model parameters are aggregated through a central server in the cloud. The ultimate global prediction model for EV RUL is formed upon the receipt and distribution of model updates. We have conducted experiments involving the extraction of various impact indicators based on authentic vehicle operational data originating from diverse geographical regions and a range of vehicle models. The main contributions are as follows: 1) Due to the limitations in obtaining comprehensive real-world usage data for EV batteries, we have devised a set of features that can be gathered and extracted directly from the EV terminal to evaluate the RUL of the battery.

2) Many prior estimation models depended on individual battery data from individual EVs for isolated model training, leading to models that could have been more reliable and lacking in generality. Additionally, gathering and transmitting heterogeneous local data for model training and updates in an online centralized manner can potentially jeopardize user privacy. We propose an RUL estimation method that utilizes a deep learning (DL) approach based on recurrent neural networks (RNN) and federated learning (FL) to predict the RUL of lithium batteries.

3) Finally, we validated the results of the model on experimental data. The model was also analyzed on actual data by comparing the mean absolute and relative errors. The comparison of the training and prediction results of the three sets of experiments shows that the federated training method achieves higher accuracy in predicting battery RUL compared to the centralized training method and another DL method, with solid training stability.

The rest of the article is organized as follows. “Related Work” discusses related work. “Methods” describes the proposed methodology. “Experimental Results” demonstrates the experimental result. “Conclusions” concludes this article and represents future work.

Related work

Classification of predictive techniques for SOH/RUL

The storage capacity and the ability for rapid charging and discharging of the battery declines with aging. This decline in battery health is most visibly evident in the reduction of available energy and power levels, alongside a decrease in overall capacity and an elevation in internal resistance. Battery state of health (SOH) is typically assessed through parameters such as battery capacity and internal resistance. In the context of this article, the defined measure of SOH is as follows:

(1) SOH=CtC0⋅100(%)

where Ct is the current capacity and C0 is the nominal capacity. In most instances, the SOH for a newly manufactured battery is established at 100%. For purely electric vehicles, where capacity demand is of primary concern, it is reasonable to assume that safety performance may decline as the battery capacity reaches 80% of its initial ability. As a result, predicting SOH can facilitate repurposing batteries for secondary use, mitigating the safety risks associated with electric vehicles.

SOH estimation methods can be divided into two main categories, namely experimental analysis methods and model-based methods, as shown in Fig. 1.

Figure 1 Classification of battery SOH estimation methods.

Specifically, as shown in Table 1, the experimental analysis method refers to analyzing the collected battery current, voltage, temperature, and other experimental data. The indirect analysis method is also divided into indirect and direct measurements depending on battery parameters. In direct measurement, specific characteristics of the battery are directly measured to determine the battery’s power. These characteristics include capacity/energy measurement, internal resistance measurement, impedance measurement, and cycle counting. The indirect analysis method is a typical multi-step derivation method, which does not directly calculate the battery capacity or internal resistance value. Still, it estimates the battery SOH by designing or measuring specific process parameters that reflect the battery capacity or internal resistance degradation, such as the capacity fade curve (Zhao et al., 2021).

Table 1 Characteristics of battery SOH estimation methods.

Methods	Advantages	Disadvantages	
Direct measurement Characterization parameters	Higher prediction accuracy;

Relatively simple, without the need for complex algorithms or models;

Easily applied in practical production and use;

	High cost of the experiment;

	
Indirect analysis Health index	Provide real-time monitoring;

Not require complex testing equipment and laboratory conditions;

	Relatively low prediction accuracy;

Not applicable to all types of batteries;

Depends on a preset model;

	
Adaptive algorithms	Higher prediction accuracy;

Used for real-time monitoring of battery life changes;

	Necessary to establish complex state estimation models;

High cost of implementation and application;

High requirements for data acquisition, transmission, and processing;

	
Data-driven methods	Higher prediction accuracy;

No need to have a deep understanding of the internal structure and characteristics of the battery, only the analysis of the operational data is required;

	Require large amounts of battery operation data;

Require a lot of data acquisition, transmission, and processing, resulting in high costs;

Require strong computing and algorithm implementation capabilities;

	

Model-based approaches require using a battery model to estimate selected battery parameters to achieve battery SOH estimation, which can be divided into adaptive state estimation algorithms and data-driven methods depending on the estimation algorithm. Adaptive algorithms generally require electrochemical models or equivalent circuit models, which are used to identify the parameters of the model and then complete the SOH prediction. These methods are distinguished by their closed-loop control and feedback mechanisms, which enable adaptive refinement of estimation outcomes based on battery voltage variations. Data-driven SOH estimation methods can predict battery life by extracting historical battery data using specific learning algorithms without detailed knowledge of the battery structure and material properties, or they can use sample entropy to assess the predictability of the battery aging time series, quantify the regularity of the data series, and analyze the battery discharge voltage data. Data-driven methods for SOH estimation can predict battery lifespan by applying specialized learning algorithms to historical battery data. These methods don’t rely on comprehensive battery structure and material properties knowledge. Alternatively, they might employ sample entropy to evaluate the predictability of battery aging time series, quantify data series regularity, and analyze battery discharge voltage data.

Constructing an accurate battery model is a challenging endeavor. Conversely, the data-driven approach relies on something other than the presence of a precise, meticulously mathematical model to depict battery aging principles and processes. Instead, it solely draws upon historical battery data, allowing for straightforward generalization across various scenarios. Therefore, the next section will focus on the progress of the deep learning-based SOH/RUL prediction solution within the data-driven approach.

Developments of deep learning

Some scholars have employed a combination of convolutional neural networks (CNNs) and long short-term memory (LSTM) networks in pertinent research regarding fault prediction through deep learning models. Qin et al. (2023) proposed using a multi-scale CNN-LSTM neural network with a denoising module for anti-noise diesel engine misfire diagnosis in their article. This method may also have application value in predicting the RUL of batteries.

Deep learning is gaining growing popularity within the realm of medical diagnosis. Some researchers combine deep learning models with clustering analysis to classify and diagnose medical images, signal data, and other data types. Mukherji et al. (2022) reviewed the current state of deep learning applications in biomedical diagnosis. It introduced an approach combining continuous clustering with deep learning models to classify and diagnose biomedical signal data effectively. This method could also hold promise for predicting the RUL of batteries.

In medical image analysis, some scholars have utilized deep learning models, combining the characteristics of CNN and LSTM to achieve the classification and diagnosis of pathological tissue images. Karimi Jafarbigloo & Danyali (2021) proposed a method based on CNN feature extraction and LSTM classification to grade nuclear atypia in breast cancer histopathological images. This approach also holds valuable applications in image processing and category, particularly for predicting the RUL of batteries.

Some scholars in mobile application development have designed efficient and time-saving applications by leveraging network services and Android application development technologies. Sarkar et al. (2015) proposed a network service-oriented Android application capable of achieving swift and effective data transmission and image processing. The application of this technology may help optimize and deploy the battery RUL prediction model on mobile devices, providing convenience for practical industrial applications.

In dialogue management optimization, some researchers have utilized deep reinforcement learning models, including experience replay-based ones, to enhance and tailor dialogue flows for optimization and personalization. Malviya et al. (2022) proposed an experience replay-based deep reinforcement learning model to optimize policy selection and decision-making processes within dialogue management. This model’s optimization applications might extend to predicting battery RUL as well. For instance, employing experience replay in data collection and preprocessing could enhance the efficiency and precision of model training for battery RUL prediction.

In network security, some scholars use machine learning methods such as decision trees, support vector machines, and neural networks to design and optimize intrusion detection systems. Hidayat, Ali & Arshad (2022) compared the effectiveness of different machine learning methods in intrusion detection systems through experiments and drew corresponding conclusions in their article. This experimental comparison method can serve as inspiration for evaluating disparities in performance among diverse deep learning models and optimization algorithms within the context of battery RUL prediction. This method can aid in selecting appropriate models and algorithms for modeling and optimization.

SOH/RUL prediction based on deep learning

In recent decades, DL has emerged as a robust tool for pattern recognition. Deep neural network architectures entail stacking multiple hidden layers, a feature that significantly boosts the learning capacity of data-driven models. Consequently, it improves accuracy and efficiency in identifying features across various domains.

Makhadmeh et al. (2021) introduced a solution termed BMO-PSPSH, which tackles the power scheduling quandary within an innovative home context, allowing for simultaneously attaining multiple objectives. The simulation results showed that BMO-PSPSH outperforms other state-of-the-art algorithms in almost all scenarios. Lin et al. (2022) presented a multi-model feature fusion approach utilizing multi-source features. Using Pearson correlation coefficients, this method initially categorized the 27 extracted health factors into three groups. Subsequently, they constructed a deep multi-model incorporating CNN, LSTM, and GraphSAGE to amalgamate the deep features into feature vectors. Ultimately, the SOH prediction was achieved through a fully connected network. A battery SOH prediction model was formulated for batteries operating under various temperatures. This model was devised by employing BP neural networks with incremental capacity analysis (Wen et al., 2022). By analyzing the correlation between IC curve characteristics and SOH, the mapping relationship between temperature and IC curve characteristics was established by the least squares method. This was done to obtain the SOH prediction model at different temperatures. Along with ICA, an online real-time correction prediction model is built, with the characteristic data continuously updated to ensure accuracy in the prediction of SOH under various aging conditions. Xia, Wang & Chen (2022) employed the fully integrated Empirical Mode Decomposition with Adaptive Noise (CEEMDAN) algorithm to decompose the raw SOH data into local fluctuations and overarching degradation trends. Subsequently, they used the GRU network and the ARIMA model to predict the abovementioned trends. Meanwhile, the second GRU algorithm is used to correct the prediction residuals of the global degenerative trend. The final SOH estimates are obtained by combining the prediction results of the above components. This method effectively addresses the negative impact of capacity regeneration and demonstrates higher accuracy and stronger robustness than other methods.

A popular approach is to use RNNs to find relationships between RUL and time series. The LSTM (Hochreiter & Schmidhuber, 1997) network is a type of RNN that can handle long-term sequences, and it has become the benchmark for recurrent networks. Therefore, LSTM and its variants are widely used in battery environments. Moreover, specific experiments have endeavored to employ convolutional neural networks (CNNs) for processing time series data or simple feedforward neural networks (FFNNs) following some form of preprocessing.

AM assigns different weights to the LSTM hidden layer to improve critical information depending on different data sets and battery capacity data with varying multipliers of discharge (Zhang et al., 2022). Additionally, Sun et al. (2022) proposed a method for predicting the SOH of lead-acid batteries using a CNN-BiLSTM-Attention model. The CNN is utilized to extract the features and reduce data dimensionality, which is then fed as input to a bidirectional LSTM (BiLSTM) that learns the time series from the local features’ time-dependent information in both directions, leading to predicting multi-step SOH of the battery. Shu et al. (2021) developed cell mean models (CMM) to predict SOH based on partial training data by combining LSTM and transfer learning (TL). They used the LSTM model to assess cell differences, applying it as a cell difference model (CDM). Based on the inconsistencies of cell SOH, they calculated the minimum CDM estimate to determine Pack SOH. The experiment resulted in a significant reduction in the amount of required training data and computational burden.

However, deep learning-based methods might not be suitable for predicting the energy of the whole EV network since they predict the energy of each EV individually. To avoid overlooking essential features such as driver behavior and traffic conditions that impact remaining battery energy, EVs should share their learned local model information instead of exclusively utilizing their dataset, leading to more accurate predictions. Therefore, using shared information or global prediction models to improve the accuracy of predicting the remaining battery energy for EVs is a challenge.

Federated learning-based energy forecasting in the electric vehicle sector

As discussed in the preceding subsection, machine learning is commonly used for energy prediction. However, traditional machine learning methods cannot train accurate energy prediction models with limited data available from a single EV. Thus, implementing federated learning can solve the issue of data silos and enhance the accuracy of predicting the remaining battery energy for electric vehicles. This technique employs shared information or global prediction models. Each end device trains a local model using its own data and shares gradient updates in horizontal federated learning. The centralized server updates the global model by aggregating the device gradients periodically. The global model is then sent back to the end devices until it achieves the desired accuracy. However, collecting and transmitting heterogeneous local data online for model training and updating can be undesirable, as it may violate users’ privacy. Furthermore, the offline anonymization of the dataset can be time-consuming and prone to errors. As a result, it is more desirable to update the model online while considering privacy concerns.

Saputra et al. (2019) proposed a federated learning approach for energy demand that enables charging stations to transmit their trained models exclusively to the charging station providers for processing. It can significantly reduce communication overhead and effectively protect the data privacy of EV users. Experimental results showed that the proposed method improves energy demand prediction accuracy by 24.63% and reduces communication overhead by 83.4% compared to other baseline machine learning algorithms.

Lu et al. (2020) proposed an asynchronous federated learning scheme that reduces transmission load and protects providers’ privacy. It also uses deep reinforcement learning for node selection to improve efficiency. Moreover, it integrates the learned model into the blockchain and performs a two-stage validation to ensure data reliability. Numerical results showed that the proposed data-sharing scheme achieves higher learning accuracy and faster convergence.

Liu (2021) proposed Fed BEV, an end-to-end federated learning framework to model the energy consumption of battery electric vehicles. The framework employs a stacked LSTM architecture to train local models and the FedAvg algorithm to aggregate them into a global model. The experimental results demonstrated that asynchronous iterations using the FedAvg algorithm can improve the predictive power of the local model.

Thorgeirsson et al. (2021) extended the federal average algorithm to train probabilistic neural networks and linear regression models in a communication-efficient and privacy-preserving manner. The study examined a network of battery electric vehicles connected to a cloud-based infrastructure that incorporates multiple relevant sources of information to forecast energy demand. To train prediction models, they utilized multi-scale regression with sensor data from the vehicle and TRDB data from the cloud. The energy demand predictions were validated with driving data, and the performance was measured using appropriate scoring rules. The study demonstrated that probabilistic forecasts outperform traditional deterministic forecasts. Additionally, the study highlighted that probabilistic energy demand forecasting benefits from a variable safety margin, resulting in improved battery energy utilization and increased effective driving range.

Saputra et al. (2020) presented a new technique for forecasting energy demand in battery electric vehicle networks through federated learning. The method involves local training of the charging transaction dataset at individual charging stations to enhance prediction precision, reduce communication overhead, and maintain information privacy. After local training, the learned model will be shared only among the charging stations without revealing their real dataset to other parties. Moreover, this article integrated federated learning with charging station clustering to optimize energy demand forecasting by reducing biased predictions caused by unbalanced features and labels.

Methods

Overall architecture

Figure 2 illustrates the proposed federated learning framework for EV RUL prediction. Local models are deployed on EVs to train local data and avoid privacy leaks during transmission by uploading model updates. Moreover, since the EVs act as sub-nodes of federated learning, they have different environments and driving habits, which makes the data distribution for training the global model more comprehensive as each node contributes additional EV driving data. Finally, the central server located in the cloud collects the parameters, receives and disseminates updates related to the model, and ultimately develops the final global prediction model for EV RUL. The overall structure consists of five steps:

Figure 2 Overall architecture.

Step 1: Collecting data on electric vehicles. Electric vehicles gather operational data through onboard devices such as built-in sensors, including vehicle ID, collection time, status updates, charging status, speed, total mileage, total voltage, total current, and other physical parameters.

Step 2: Data preprocessing. The data processor of the electric vehicle performs data cleaning (deduplication), calculates simple features, and extracts features from the physical parameters of vehicle driving collected during Step 1.

Step 3: Build the initial model. After extracting features, the input ones are filtered. The initial global model is designed on the central cloud server, which includes the model inputs/outputs, model structure, and loss function.

Step 4: Begin the federated learning process. The central cloud server transmits the initial global model to every end node (in this case, an electric vehicle). The end nodes receive the global model, update their local model with the local data, and upload the model update information to the central cloud server. The server aggregates the parameters and updates the global model, then sends the model parameters again and repeats this process until the global model reaches a predefined threshold value.

Step 5: The cloud server shares the final global model with all end EVs when federated learning is complete. Then, each EV predicts its own remaining battery life cycle, considering its historical driving data and current driving conditions.

Data preprocessing

Data cleaning

Technical defects in the sensors and complex operating conditions can sometimes lead to signal delays, false positives, or even data loss during GPS data transmission, which can cause anomalies in the collected data. Therefore, data cleaning is required. In case of duplicate data, all those data records except one are eliminated, and a single record is retained. In cases of missing data, data padding is applied. For instance, if the mileage values are absent for a segment, they are replaced with the mileage values of that particular segment. If a segment’s mileage values are missing, they are filled with the mileage values at the end of the previous segment.

Sliding window to calculate the battery capacity

Analyzing the factors that influence battery RUL is necessary for accurate prediction. In current battery research, state of charge (SOC) is the most common feature, which reflects the battery’s remaining capacity and decays with the number of cycles. It is defined numerically as the ratio of remaining to battery capacity. Therefore, it has a strong correlation with the battery RUL. This article uses the ampere-time integration method to calculate battery capacity with the following equation:

(2) C=∫I¯dtΔSOC

where C is the calculated capacity, I¯ is the current fragment current mean value. ΔSOC is the difference between the maximum SOC and the minimum SOC within the fragment.

As shown in Fig. 3, the battery ID is the same for a charging process in the data set, and the first data point is the starting point of that process. The battery capacity of this window is calculated by applying the ampere-time integration method, and the window is slid by one step until the end of charging (i.e., the last data point). The sliding window size is set to 60 records, meaning one unit per 60 records. Then, the battery capacity of this process is calculated by estimating the average capacity of all windows.

Figure 3 Sliding window to calculate battery capacity.

Feature extraction

To predict the accurate RUL of a battery, it is necessary to collectively consider environmental, vehicle operating, and historical factors. As shown in Table 2, the temperature of the external environment impacts the electrochemical reactions inside the battery, which in turn affects its charging and discharging performance. There is a significant difference in the vehicle’s operating characteristics between emergency braking and normal driving conditions. As batteries age, their remaining life cycle decreases.

Table 2 Features extracted.

Type	Feature name	Data type	Explanation	
Environment feature	mon_a_temp	Float	Average temperature by month	
Vehicle operation features	mon_day	Int	Total driving days by month	
	mon_mile	Float	Total driving mileage by month	
	mon_cycle	Int	Total charging and discharging cycles by month	
	mon_acc_Time	Int	Total acceleration time by month	
	mon_acc_time	Int	Total acceleration times by month	
Battery features	mon_a_cap	Float	Average capacitance by month	
	mon_a_R	Float	Average resistance by month	
	mon_a_I	Float	Average current by month	
	mon_a_V	Float	Average voltage by month	
	mon_use_soc	Float	Total electricity consumption by month	
	mon_a_V_diff	Float	Average voltage range by month	
	mon_a_temp_diff	Float	Average temperature range by month	
	soc	Float	State of charge	
Historical feature	a_cycle	Int	Total cycles	
	a_days	Int	Total driving days	
	a_mileage	Float	Total mileage	

We use an RNN model based on time series, which splits the input by time. Considering the spatial and temporal distribution of electric vehicle operation, the frequency of car use is higher on holidays than on weekdays. Moreover, the battery wear and tear is increased. Therefore, this article extracts each feature by month.

Model construction

Before presenting the model construction, the following definitions of features are given: 1) F=(f1,f2,⋯,f16): A row vector of dimension 1 × A row vector of dimension 16, representing the ith feature vector of a given cell.

2) EVi=(F1,F2,⋯,Fn)−1: denotes the eigenmatrix consisting of all eigenvectors of cell i.

3) FM=(EV1,EV2,…,EVn)−1: The set of all electric vehicle battery feature matrices.

Analysis of the problem

The RUL of a battery refers to the point where its performance or health has declined to the extent that it can no longer sustain the equipment’s operation under specific charge and discharge conditions or after it has undergone a specified number of charge and discharge cycles. The SOH of a battery usually refers to the parameters that characterize the battery’s health. These parameters are also known as health factors. This article calculates SOH using the capacity measurement method, which accurately measures the current maximum available capacity of the battery as a percentage of its rated capacity. The capacity measurement method uses capacity as a health factor, and the formula for defining SOH is as follows:

(3) CSOH=CMCN×100%

where CSOH is the SOH as defined by the capacitance method. CM is the current stable capacity of the battery. CN is the rated capacity. The RUL prediction is an assessment of the remaining life cycle of the battery before failure, generally defined as a battery failure at 80% SOH, and is given by:

(4) RUL=TSOH80%−TNow

where RUL represents the remaining life cycle of the battery. TSOH80% represents the time at which the battery SOH reaches 80%. TNow represents the time under the current SOH of the battery.

RNN models

Data with time-series characteristics can be handled by RNNs using the information from the hidden layer neurons of the network. Specifically, in the RUL prediction scenario, the battery aging data is extracted from the feature parameters and fed into the RNN for training. This structure fits the battery decline curve well and accurately predicts the battery’s remaining life.

First, the feature matrix FM was normalized. Then the FMs are cropped according to different lengths to form a set H={EV1,EV2,…,EVn} containing multiple subsets of FMs as input to the model. The shape of each FM subset is 16 * N. N is the batch size during training. As shown in Fig. 4, the prediction of RUL is achieved by calculating the loss of each vector input for each FM subset.

Figure 4 Model input & output.

As shown in Fig. 5, the model consists of an input, hidden, and output layer. The number of neurons in each fully connected layer is set as low as possible to keep the model lightweight. The hidden layer has five layers of simple recurrent neural networks with 64,32,16,8,4 neurons and two thoroughly combined layers.

Figure 5 RNN model structure.

We use mean squared error (MSE) and mean absolute error (MAE) to assess the difference between the predicted and actual values of the model using the following formula:

(5) MSE=1m∑i=1m(y^i−yi)2

(6) MAE=1m∑i=1m|y^i−yi|

where yi denotes the actual value of SOH, yi^ is its corresponding predicted value, and m represents the feature dimension.

Federated learning

The training process for federated learning consists of two parts: global model training and local model training. Clients update their local models based on their individual data and transmit the updates to the central server. Then, the central server aggregates the updates to calculate a modified global model.

Local training of the model consists of four steps: Step1: The initial global model is received from the incoming central server.

Step2: Collecting multiple EVs in a trusted vehicle network to form an ensemble H={EV1,EV2,…,EVn} training dataset.

Step3: EVi is taken from the set FM at each training session, and the model is trained using the gradient descent algorithm until all the EVs in the set FM are trained.

Step4: Receive the updated model parameters from the central server after uploading the weights and biases of the trained model. Repeat steps 3 and 4 several times until the global node converges globally.

Global training consists of three steps: Step1: Design the model and distributes it to each node.

Step2: Collect model parameters and losses for local training at each node.

Step3: The parameters and losses of each node are aggregated and resent to each node. Repeat steps 2) and 3) several times until the global loss reaches the set convergence threshold.

Step4: Considering the different data volumes of the sub-nodes when aggregating the model globally, the ratio of each node’s data volume to the total data volume is used as the weight for aggregation.

Experimental setup

To validate the performance of the proposed method, we conducted experiments with data collected from automotive sensors. We used 124 batteries, each with several charge and discharge cycles. Our focus was on two aspects: (1) the stability of federated training; and (2) the accuracy of the model’s predictions.

Figure 6 shows the experimental procedure, including data preprocessing, data splitting, and comparison experiments. The 124-cell data collected by the sensors was cleaned and normalized. The dataset required for the experiments was formed by extracting each cell’s data over time. The dataset was then split into a training set and a validation set by battery, and a bunch of comparison experiments were conducted. The federated training experiments divided the training data into five nodes, and the centralized training experiments used the whole training data. Finally, we analyzed the results of the two experiments.

Figure 6 Overall experimental process.

Experimental environment and dataset

The experiments were conducted on a computer system that included an Intel(R) Core(TM) i5-8250U CPU processor and an Intel(R) UHD Graphics 620 graphics card operating on the Windows 10 platform. The TensorFlow deep learning framework, version 1.10.0, and Python programming language, version 3.6.2, were utilized. The dataset employed for the experiments was acquired from a published study (Severson et al., 2019). The dataset was collected from 124 commercial lithium iron phosphate/graphite batteries subjected to fast-charging cycles, with cycle lives ranging from 150 to 2,300 cycles.

Data preprocessing

First, the data has many spikes, which may represent some errors in the sensor readings or other anomalous data that need to be cleaned up. Here the data is processed using an exponential moving average to remove outliers and ‘smooth out’ problematic curves. Secondly, the features in the dataset have different value ranges. Hence, the data needs to be normalized to a specific interval ([−1,1]) to make the features comparable, eliminate the undesirable effects caused by odd sample data, speed up gradient descent to find the optimal solution and improve accuracy. Third, the battery IDs in the dataset are the same for a battery with multiple charge/discharge cycles. We obtain 60 “windows” of 100 cycles from the first 160 cycles of each battery as all the data for that battery, treating them as many time series, each with many features, and maintaining the temporal order of the series. Finally, there are 124 cells in the dataset, each with 60 sliding windows. This results in a total of 7,440-time data series, each containing 100 data cycles, as shown in Table 3.

Table 3 Dataset description.

Data items	Numerical values	
Number of batteries	124	
Sliding window size	100	
Number of acquisition windows per cell	60	
Total number of sequences	7,440	
Total number of cycles	744,000	
Number of features	16	

Data splitting

Due to the large number of batteries and the small amount of data that can be obtained for each battery, we merged the data by battery and split it into a training and validation set. The data from 120 batteries, from batteries 0 to 119, were used as the training set. The data from four batteries, 120 to 123, were used as the validation set. It means that the training set has 7,260 time series data, which is 726,000 data cycles. The validation set has 180 time series data, which is 18,000 data cycles. Further, according to the number of sub-nodes in the federated training, the training set is split into five parts by battery. We combine every 24 batteries into one battery, equivalent to a training set of five batteries in the federated training, and each sub-node training set has 1,440 time series data and 144,000 cycle data.

Parameter setup

We designed a series of comparative experiments comprising three components to assess our concerns regarding the stability of federated training and the accuracy of model predictions. Specifically, we conducted experiments utilizing the RNN federated training method, the RNN centralized training method, and the CNN-ATSLSTM method proposed by Li et al. (2022) in their study, respectively, for RUL prediction. The respective parameter configurations for these experiments are presented in Table 4.

Table 4 Experimental parameter settings.

Parameter name	Federated training	Centralized training	CNN-ATSLSTM	
Number of nodes	5	–	–	
Data volume by node	1,440	–	–	
Total data volume	7,200	7,260	7,260	
Local training rounds	100	1,000	1,000	
Global training rounds	10	–	–	
Optimizer	Adam	Adam	Adam	
Learning rate	0.0005	0.0005	0.0005	
Data batch size	64	64	64	

Federated training: The number of nodes in the federated training experiment was restricted to five; each node used 24 batteries of data as the training data set, containing 1,440 time series data, i.e., 144,000 cycles of data; the total data volume of five nodes was 7,200; the number of local training rounds was limited to 100; the number of global training rounds was 10 rounds, each node would train 1,000 rounds. The optimizer is Adam, the learning rate is set to 0.0005, and the data batch size is 64.

Centralized training: The amount of data in the centralized training experiment is 7,260 time series data, i.e., 726,000 cycles of data; the local training rounds are set to 1,000. The optimizer is Adam; the learning rate is set to 0.0005, and the data batch size is 64.

CNN-ATSLSTM: The training set data volume is 7,260 time series data, i.e., 726,000 cycles of data; the number of local training rounds epochs is set to 1,000; the optimizer is Adam; the learning rate is set to 0.0005, and the data batch size is 64.

Evaluation metrics

The evaluation metrics used in the experimental component are:

Mean Squared Error Loss (MSE Loss) measures how bad a neural network’s performance is. It is the average of the sum of the squares of the differences between the predicted and target values, calculated as:

(7) MSELoss={mean[(yi−y^i)2],reduction=meansum[(yi−y^i)2],reduction=sum}

Mean Absolute Error (MAE), which is the average of the absolute errors, better reflects the actual situation of the forecast value error and is calculated as:

(8) MAE=1m∑i=1m|yi−y^i|

Residuals, the difference between the actual and estimated values, are used to measure the difference between the predicted and true values and are calculated as:

(9) Residuals=yi−y^i

In the above equation, yi^ is the true value of the target, and yi is the predicted value.

Experimental results

Comparison of training stability

Figure 7 displays the trend of MSE Loss change for the three experimental sets, where RNN-FL_loss and RNN-FL_val_loss denote the MSE Loss of federated training. It can be observed that the MSE Loss of all three models converges to the lowest value within 300 epochs. As the Loss value decreases, the val_Loss value also decreases, indicating standard model training. The federated training and CNN-ATSLSTM curves exhibit relatively smooth trends with minor fluctuations in values, indicating strong training stability. At the beginning of training, the MSE Loss of federated training converges faster than that of CNN-ATSLSTM. The specific MSE Losses of each epoch are presented in Table 5. The overall MSE Loss of federated training is lower than the other two models, and the final convergence at the end of training attains values of 231.7720 and 99.5836, respectively, which are lower than the final values of centralized training and CNN-ATSLSTM. Thus, federated training exhibits more substantial stability with lower loss and superior training results.

Figure 7 Trends in MSELoss.

Table 5 MSE Loss values.

	Centralized training	CNN-ATSLSTM	Federated training	
Epochs	Loss	val_Loss	Loss	val_Loss	Loss	val_Loss	
0	491,646.0032	185,134.0506	486,878.0032	483,986.0506	486,979.0536	209,103.2877	
1	134,666.7335	133,816.1474	479,478.7335	476,722.1474	297,430.8339	247,654.1310	
2	132,730.9633	145,277.1068	471,960.9633	468,408.1068	138,228.1763	131,975.9990	
…	…	…	…	…	…	…	
995	1,311.0405	1,640.3068	635.4150	134.6640	246.9252	102.1924	
996	1,735.1783	1,098.3275	636.7056	114.3281	243.9156	93.9880	
997	1,829.7231	1,985.8028	608.3708	95.4834	233.9361	105.1113	
998	1,555.3295	1,735.3379	601.0994	127.1374	247.6579	112.2270	
999	1,334.7052	812.2385	627.3907	99.5114	231.7720	99.5836	

Comparison of prediction accuracy

MAE

Figure 8 shows the MAE variation trend for the three experiment sets. RNN-FL_MAE and RNN-FL_val_MAE indicate the MAE of federated training. The MAE and val_MAE of all three models converge at a faster rate. The MAE of the centralized training model is larger and exhibits more fluctuations. In contrast, the MAE of federated training and CNN-ATSLSTM drop smoothly to a lower value and fluctuate steadily in a smaller range. Furthermore, the MAE of federated training converges first. It indicates that the training error of the federated training model is smaller than that of the centralized training model and can reach about the same accuracy as that of the CNN-ATSLSTM model.

Figure 8 MAE change curve.

Model prediction results

The trained model was utilized to predict the battery RUL on the test set data, and the accuracy of the prediction results was analyzed. Figure 9 compares the predicted and actual values of the three models using the first 100 data of the test set for analysis. The solid blue line represents the RUL values predicted by the models, and the dashed orange line represents the true RUL label values. The figure shows that the prediction results of all three sets of experiments are in good agreement with the actual values. However, the RUL predictions of the centralized training model deviate significantly from the real values in more parts, and a small portion of the CNN-ATSLSTM model also had inaccurate predictions. In contrast, the prediction values of the federated training model were highly consistent with the actual values. The federated training model demonstrates more accurate prediction results.

Figure 9 Comparison between model predictions and true values.

Residual analysis

Figure 10 shows the range and distribution of residuals values for the predicted results. It includes univariate distribution plots (histograms and kernel density plots) of residuals and residuals values for each model. The residuals of the centralized training model are mainly concentrated in the (−50, 50) range, while the CNN-ATSLSTM model and the federated training model are primarily in the (−20, 20) and the (−10, 10) ranges, respectively. All three models have approximately normally distributed residuals, indicating accurate data predictions. However, the overall values of the residuals of the federated training model are smaller than those of the other two models. Moreover, the Residuals statistical analysis in Table 6 shows that the federated training model has better Residuals mean, standard deviation, and other statistical values. Therefore, the federated training model is more accurate in predicting RUL than the centralized training model and the CNN-ATSLSTM model.

Figure 10 Residuals comparison.

Table 6 Statistical analysis of residuals.

Statistical quantities	Centralized training	CNN-ATSLSTM	Federated training	
Count	1,513.000000	1,513.000000	1,513.000000	
Mean	12.437315	1.963115	1.626886	
Std	25.424813	9.350229	4.636589	
Min	−108.880493	−46.157959	−14.993921	
25%	−1.657349	−5.813110	−1.462158	
50%	11.755493	1.190369	0.871124	
75%	26.967987	6.100830	4.108337	
Max	140.454712	50.402954	24.906982	

Conclusions

The proposed RNN-based federated learning method for RUL prediction provides a promising approach for addressing privacy concerns while achieving high prediction accuracy. The privacy of user battery data is protected by partitioning the models into local and global models and uploading only model updates during training. Furthermore, using battery data from different sub-nodes to train the global model results in a complete data distribution compared to centralized training methods and other existing RUL prediction methods. The comparison of the training and prediction results of the three sets of experiments shows that the federated training method achieves higher accuracy in predicting battery RUL compared to centralized training and CNN-ATSLSTM methods, with solid training stability. Overall, the proposed method protects the privacy of user battery data and achieves good training stability and higher prediction accuracy, making it a promising approach for RUL prediction in the context of battery management systems.

Additional Information and Declarations

Competing Interests

Author Contributions

Data Availability

The authors declare that they have no competing interests.

Xuejiao Chen conceived and designed the experiments, performed the experiments, analyzed the data, performed the computation work, prepared figures and/or tables, authored or reviewed drafts of the article, and approved the final draft.

Zhaonan Chen analyzed the data, performed the computation work, authored or reviewed drafts of the article, and approved the final draft.

Mu Zhang conceived and designed the experiments, performed the experiments, analyzed the data, authored or reviewed drafts of the article, and approved the final draft.

Zixuan Wang conceived and designed the experiments, performed the computation work, prepared figures and/or tables, and approved the final draft.

Minyao Liu performed the experiments, prepared figures and/or tables, and approved the final draft.

Mengyi Fu performed the experiments, prepared figures and/or tables, and approved the final draft.

Pan Wang analyzed the data, prepared figures and/or tables, authored or reviewed drafts of the article, and approved the final draft.

The following information was supplied regarding data availability:

The code is available at GitHub and Zenodo:

- https://github.com/PrinceXuan1/FL-Battery-RUL/releases/tag/dataset

- Wang Zixuan. (2023). FL-Battery-RUL. Zenodo. https://doi.org/10.5281/zenodo.8219784

The data is available at GitHub:

- https://github.com/PrinceXuan1/FL-Battery-RUL/releases/tag/dataset

- Wang Zixuan. (2023). FL-Battery-RUL-dataset [Data set]. Zenodo. https://doi.org/10.5281/zenodo.8379081

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
