# Peer review of "A remaining useful life estimation method based on long short-term memory and federated learning for electric vehicles in smart cities"

_PeerJ Computer Science, doi:10.7717/peerj-cs.1652_

## Round 0.1 · original submission · Major Revisions

Hi,

Both the reviewers raised major concerns, which should be justified with evidence from the experimental results.

/Khursheed

Reviewer 1 ·

Basic reporting

The article presents a research study focused on developing a method for accurately estimating the Remaining Useful Life (RUL) of lithium battery packs used in Electric Vehicles (EVs). The authors highlight the importance of precise RUL estimation for enhancing personal safety and facilitating secondary trading and recycling in other industries. The abstract provides a clear summary of the research problem and the purpose of the study. The methods and results are briefly described, but not in enough detail to fully understand the study. The article also mentions the challenges faced by existing methods and how the proposed method addresses these challenges but without much detail. Discussion on related studies missed the use of heuristic (nature-inspired) optimisation methods such presented in Makhadmeh et al. (2021). Smart home battery for the multi-objective power scheduling problem in a smart home using grey wolf optimizer. Electronics, 10(4), 1-35.

Experimental design

The authors describe the use of a Deep Learning (DL) approach based on Long Short-Term Memory (LSTM) and Federated Learning (FL) to predict the RUL of lithium batteries. The authors claim that they avoided using unmeasurable variables as input and that the FL framework was used to protect users' battery data privacy. The results are verified on experimental data and analyzed on real data using mean absolute error and mean relative error metrics.

Validity of the findings

Thearticle does not provide sufficient information to assess the validity of the findings. The results are described as meeting the needs of daily use, but without more detail it is difficult to determine how well the method performed. The comparison with other methods is not discussed, so it is unclear how the proposed method compares to existing methods.

Additional comments

More detail is needed to fully evaluate the validity of the findings. The use of DL and FL is promising for the RUL estimation of lithium batteries, but the article does not provide enough information to determine how well the method performs when compared to existing methods. Further analysis and discussion of the results, limitations, and implications of the study would also be useful.

Reviewer 2 ·

Basic reporting

In this paper, the authors have proposed an RUL estimation method utilizing a Deep Learning (DL) approach based on Long Short-Term Memory (LSTM) and Federated Learning (FL) to predict the RUL of lithium batteries. They have avoided using unmeasurable variables as input and designed the estimation model based on LSTM by taking advantage of predicting the time series data. Also, the authors have applied the FL framework to train the model to protect users' battery data privacy.

Experimental design

The experimental design is convincing.

Validity of the findings

The authors have not discussed the results properly.

Additional comments

1. The Abstract must be very precise.
2. The Introduction section is very poor. In a research article, the introduction section must be very strong with the motivations of this paper, which is missing in this paper. Moreover, the disadvantages of the existing schemes must be discussed to motivate this new work.
3. The point-wise contributions are not convincing. The last paragraph of the Introduction should be the structure of the paper.
4. The Related Work section is poor. The authors must include some more schemes. Also, the following papers must be cited to improve this section, as well as the Reference section:
a) Anti-noise diesel engine misfire diagnosis using a multi-scale CNN-LSTM neural network with denoising module
b) Recent landscape of deep learning intervention and consecutive clustering on biomedical diagnosis
c) An efficient and time saving web service based android application
d) Nuclear atypia grading in breast cancer histopathological images based on CNN feature extraction and LSTM classification
e) Experience replay-based deep reinforcement learning for dialogue management optimisation
f) Machine learning based intrusion detection system: An experimental comparison
g) IFODPSO-based multi-level image segmentation scheme aided with Masi entropy
5. In Related Work, a table can be given to summarize the entire section.
6. What is the use of SOC in the proposed scheme?
7. How the performance is increased?
8. In Eq. (5), how the value of m is decided?
9. Cleaning stage is completely unclear.
10. Which entities are involved in battery management?
11. How Eq. (6) improves the performance of the model?
12. In Experimental Results, add the “Experimental Environment” subsection.
13. What is the source of dataset? Whether it is authentic or not? Mention clearly.
14. How the results of Figure 8 are generated?
15. Technical details about results are missing.
16. The caption style of the multi-figure is wrong.
17. What is the novelty of this work? It is hard to identify from the current version of this paper.
18. Key terms of the equations must be defined.
19. Use a well-known software to draw the diagrams of the results section.
20. The organization of the paper must be improved. The paper must be formatted properly.
21. Improve the English language.
22. The Reference section must be improved significantly.
23. Add section numbering in the revised paper.

---

## Round 0.2 · accepted · Accept

Both reviewers are satisfied with the revised version of the manuscript.

Reviewer 1 ·

Basic reporting

The language is good.
Literature references are sufficient.
Figures are well execute.

Experimental design

The experiments were designed and executed correctly.

Validity of the findings

The findings are valid.
The conclusions are suitable.

Additional comments

I have no further comments. The manuscript can be accepted for publication.

Reviewer 2 ·

Basic reporting

The authors have addressed all the comments.

Experimental design

No comment.

Validity of the findings

No comment.

Additional comments

No comment.